

# On revealing the gene targets of Ebola virus microRNAs involved in the human skin microbiome

Pei-Chun Hsu[1], Bin-Hao Chiou[2] and Chun-Ming Huang[3]

[1] Department of Biomedical Electronics and Bioinformatics, National Taiwan University, Taipei, Taiwan
[2] Department of Systems Biology and Bioinformatics, National Central University, Jhongli, Taiwan
[3] Department of Medicine, Division of Dermatology, University of California, San Diego, CA, United States of America

## ABSTRACT

Ebola virus, a negative-sense single-stranded RNA virus, causes severe viral hemorrhagic fever and has a high mortality rate. Histopathological and immunopathological analyses of Ebola virus have revealed that histopathological changes in skin tissue are associated with various degrees of endothelial cell swelling and necrosis. The interactions of microbes within or on a host are a crucial for the skin immune shield. The discovery of microRNAs (miRNAs) in Ebola virus implies that immune escape, endothelial cell rupture, and tissue dissolution during Ebola virus infection are a result of the effects of Ebola virus miRNAs. Keratinocytes obtained from normal skin can attach and spread through expression of the thrombospondin family of proteins, playing a role in initiation of cell-mediated immune responses in the skin. Several miRNAs have been shown to bind the 3′ untranslated region of thrombospondin mRNA, thereby controlling its stability and translational activity. In this study, we discovered short RNA sequences that may act as miRNAs from *Propionibacterium acnes* using a practical workflow of bioinformatics methods. Subsequently, we deciphered the common target gene. These RNA sequences tended to bind to the same thrombospondin protein, THSD4, emphasizing the potential importance of the synergistic binding of miRNAs from Ebola virus, *Propionibacterium acnes*, and humans to the target. These results provide important insights into the molecular mechanisms of thrombospondin proteins and miRNAs in Ebola virus infection.

# BACKGROUND

## Ebola virus (EBOV) infections and host responses

EBOV is a negative-sense single-stranded RNA virus; five species have been identified, including four that primarily affect humans. EBOV can cause sporadic outbreaks of hemorrhagic fever with high fatality rates in Central Africa and Southeast Asia.

EBOV is primarily transmitted from human to human by direct contact with blood or tissue of infected patients. The virus infects host cells by destroying the protein synthesis systems and immune defenses (*Beer, Kurth & Bukreyev, 1999*) at an unexpected high

Corresponding author
Pei-Chun Hsu,
d98945017@ntu.edu.tw

replication rate. Host immune responses against EBOV and direct infection of monocytes and macrophages result in the release of cytokines. The immune responses associated with inflammation and fever lead to Ebola hemorrhagic fever (EHF) (*Yang et al., 2000*).

Histopathological and immunopathological studies of EBOV have revealed that EBOV infection is unconventional. Histopathological changes in the skin tissue consist mainly of various degrees of endothelial cell swelling and necrosis (*Zaki et al., 1999*). Moreover, in some cases, EBOV infections cause moderate to severe clinical manifestations.

## Bioinformatics analysis and prediction of potential EBOV microRNAs (miRNAs)

Recently, the genome-wide screening of samples from the 2014 EBOV outbreak demonstrated the presence of putative viral miRNAs in EBOV, implying that EVOB miRNAs may target specific genes. The interactions of miRNA and target genes during EBOV infection may regulate endothelial cell rupture and tissue dissolution related to immune escape of host defense mechanisms (*Teng et al., 2015*).

miRNAs are small regulatory noncoding RNAs (18–23 nucleotides in length) that are processed from long precursor transcripts (pre-miRNAs). Cellular miRNAs are highly conserved and naturally bind to complementary sequences in the 3′-untranslated region (3′-UTR) of target mRNA, thereby downregulating the production of the corresponding protein (*Filipowicz, Bhattacharyya & Sonenberg, 2008*; *Siomi & Siomi, 2010*). One of the most important aspects of miRNA function is their binding to mRNAs at nucleotide positions 2–7, representing the "seed region" or "seed sequence". Although the base pairing of a miRNA and its target mRNA does not have to be exact, the seed sequence must be perfectly complementary (*Hibio et al., 2012*).

## Interactions of microbes and opportunistic pathogens

Opportunistic pathogens, such as *Propionibacterium acnes*, a gram-positive anaerobic predominant bacterium in the skin, either within or on a host, are a crucial aspect of the skin's immune shield (*Schommer & Gallo, 2013*). Microbial colonization on the skin, also known as the skin microbiome, strengthens the skin's defense against potentially pathogenic organisms. Keratinocytes obtained from normal skin attach and spread through binding to the thrombospondin (TSP, also referred to as THSD) family of proteins; this interaction may play a role in initiating cell-mediated immune responses in the skin by releasing cytokines and stimulating the expression of TSP proteins (*Varani et al., 1988*) to facilitate the movement of immune competent cells (*Baker et al., 2003*). TSP promotes keratinocyte attachment and spreading and may play an important role in maintaining the normal growth of the basal cell layer. To date, several miRNAs have been observed to bind the 3′ UTR of the TSP mRNA, as well as controlling its stability and translational activity (*Bartel, 2009*).

The human genome contains more than 1,000 miRNA genes that have been either predicted or experimentally observed to play critical roles in normal cellular functions, such as maintaining homeostasis and regulating or modulating viral and cellular gene expression. Thus, miRNAs constitute one of the most abundant classes of gene-regulating molecules in animals.
### A new approach to identification of the mechanism of EBOV infection

Currently, no specific therapies have been shown to be effective for the treatment of EHF. Thus, only supportive therapies are typically given to attempt to overcome the infection; however, the fatality rate remains high. EBOV can spread through direct skin contact. Recent studies have proposed that EBOV infection is a result of the action of EBOV miRNAs (*Liang et al., 2014*). As previously mentioned, microbes in the skin microbiome, such as *P. acnes*, potentially play a vital role in protecting the host.

In this study, we developed a bioinformatics approach based on sequence alignment to detect short RNA fragments in *P. acnes*, similar to viral miRNAs in EBOV, and to determine the mechanism of EBOV infection.

## METHODS

### Identification of short RNA fragments in *P. acnes*

The whole genome of *P. acnes* KPA171202 (*Bruggemann et al., 2004*) was retrieved from the National Center for Biotechnology Information (NCBI) website (http://www.ncbi.nlm.nih.gov/nuccore/NC_006085.1). The predicted mature miRNAs in EBOV were used as the query sequence.

The Basic Local Alignment Search Tool (BLAST) Two sequence program was employed to search against the entire *P. acnes* genome at the NCBI (http://blast.ncbi.nlm.nih.gov/). The specialized BLAST program, called "Align Two Sequences" (bl2seq), was selected to perform pairwise local alignments of whole-genome sequences with the following parameter settings: word length, 11; *E*-value cutoff, 10. The RNA sequences used as the query in BLAST were as follow: *EBV-miR-1-5p*, 5'-AAAAAGUCCAUAAUGCUGGGGA-3'; *EBV-miR-1-3p*, 5'-GCCACCAUAGGACUUUUUCAAU-3'; *EBV-miR-2-3p*, 5'-UUAUCCUUCUUGAAUCCUGAGA-3'.

### Target gene prediction

The target genes of RNA fragments were predicted by TargetScan online software (release 6.0; http://www.targetscan.org/) (*Agarwal et al., 2015*). MiRNAs regulate protein coding gene expression by binding to the 3' UTR, and TargetScan is specifically designed for predicting such interactions. Thus, we used TargetScan to predict the binding of the seed regions of miRNAs. We employed customized TargetScanHuman software (version 5.2; http://www.targetscan.org/vert_50/seedmatch.html) to search for the predicted target genes of small RNA fragments. This customized software was specifically developed to predict the biological targets of RNA fragments by searching for the presence of 7–8 nucleotides that match the seed region of each RNA fragment. For example, in *P. acnes* RNA fragments, a short nucleotide sequence (nucleotides 1–7 from the 5' end acts as the seed sequence. The RNA fragments applied for these analyses are shown in Table 1 and were derived from three sources: EBOV, *P. acnes*, and humans. As a result, the three groups of gene lists were separated into EBOV miRNA, *P. acnes* short RNA sequences, and human *miR-1248* based on queries in miRBase. We cross-matched these three gene lists to find common target genes.

**Table 1  Short RNA sequences employed in this study.**

| RNA sequence id | Source | RNA sequence (5′→ 3′) | Length (bp) |
|---|---|---|---|
| *EBV-miR-1-5p* | Ebola virus | **AAAAAGUC**CAUAAUGCUGGGGA | 22 |
| *EBV-miR-1-3p* | Ebola virus | **GCCACCAU**AGGACUUUUUCAAU | 22 |
| *EBV-miR-2-3p* | Ebola virus | **UUAUCCUU**CUUGAAUCCUGAGA | 22 |
| *PA-miR-1-5p-1* | P. acnes | **GCAUUAU**GGACU | 12 |
| *PA-miR-1-5p-2* | P. acnes | **AUCAUGG**ACUUUUU | 14 |
| *PA-miR-1-5p-3* | P. acnes | **AAGUCCA**UAAU | 11 |
| *PA-miR-1-5p-4* | P. acnes | **CCCCAGC**AUUA | 11 |
| *PA-miR-1-3p-1* | P. acnes | **CCUAUGG**UGGC | 11 |
| *PA-miR-1-3p-2* | P. acnes | **GCCACCA**UAGG | 11 |
| *PA-miR-1-3p-3* | P. acnes | **GUCCUAU**GGUG | 11 |
| *PA-miR-1-3p-4* | P. acnes | **AUAGGAC**UUUU | 11 |
| *PA-miR-1-3p-5* | P. acnes | **AGGACUU**UUUCGAU | 14 |
| *PA-miR-1-3p-6* | P. acnes | **UGAAAAA**GUCC | 11 |
| *PA-miR-1-3p-7* | P. acnes | **ACUUUUU**CAAU | 11 |
| *PA-miR-2-3p-1* | P. acnes | **UAUCCUU**CUUGA | 12 |
| *PA-miR-2-3p-2* | P. acnes | **AUCCUUC**UUGA | 11 |
| *PA-miR-2-3p-3* | P. acnes | **UCAAGAA**GGAU | 11 |
| *PA-miR-2-3p-4* | P. acnes | **UCCUUCU**UGAA | 11 |
| *PA-miR-2-3p-5* | P. acnes | **CCUUCUU**GAAU | 11 |
| *hsa-miR-1248* | Human | A**CCUUCUU**GUAUAAGCACUGUGCUAAA | 27 |

Notes.

P. acnes, *Propionibacterium acnes* KPA171202. The target genes of RNA fragments were predicted by TargetScanHuman online software. The seven nucleotides highlighted in bold for each sequence are the seed sequences.

## Analysis of *THSD4* mRNA expression

To explore the regulation of gene expression by miRNAs, it is necessary to identify the target genes and regulatory effects of each miRNA. In order to further demonstrate the regulatory role of THSD4, human embryonic kidney 293T (HEK293T) cells (American Type Culture Collection, Manassas, VA, USA) were transfected with short RNA sequences using siTran transfection reagent (Origene) according to the manufacturer's protocol. Cells were then incubated for 48 h, and the expression level of *THSD4* mRNA was quantified by real-time quantitative polymerase chain reaction (RT-qPCR). The fold change in mRNA was computed based on the change in threshold cycle (Ct) and represented by $2^{-\Delta\Delta Ct}$ values.

## RESULTS AND DISCUSSION

### Prediction of P. acnes small RNA fragments from EBOV miRNAs

As the availability of sequence and biological information has increased, the fields of bioinformatics and computational biology have begun to play major roles in the study of fundamental biomedical problems. Biologists face challenges associated with vast amounts of data produced by experiments; analysis of these data may reveal previously unknown relationships with respect to the functions of genes and proteins. Moreover, the development of appropriate databases could allow biologists to integrate information by
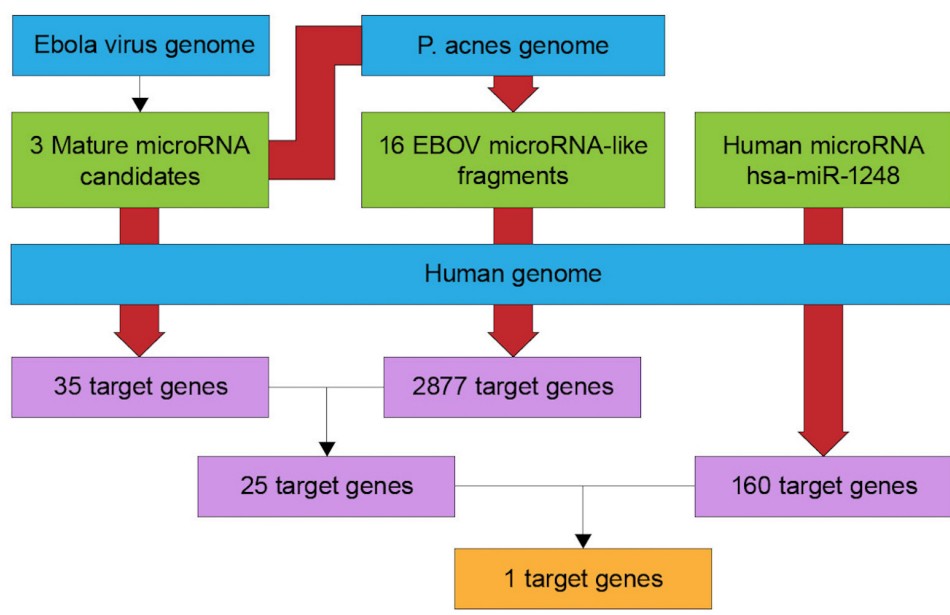

**Figure 1  Flowchart of target gene prediction.** The BLAST Two sequence program was employed to predict short RNA sequences from *P. acnes*. All target genes were predicted using the TargetScan web service. Each red arrow represents a BLAST process.

themselves rather than relying on programmers. Here, we created a workflow for retrieving entries from databases and annotating a collection of publicly available sequences (Fig. 1).

MiRNAs are thought to be expressed primarily in eukaryotes. Recently, miRNAs have also been shown to be employed by viruses in order to regulate the expression of viral genes, host genes, or both (*Grice & Segre, 2011*). EBOV miRNAs can be generated through cellular miRNA processing machinery. Three mature miRNAs have been identified from EBOV (*Liang et al., 2014*).

By mapping the clean reads of EBOV miRNAs onto *P. acnes* and the human genome, 16 small RNA fragments from *P. acnes* and one human miRNA were detected in this study. We obtained only sequences with 100% similarity, as well as several mismatches. The lengths of the identified small RNA sequences were approximately 11–13 nt each. The newly identified RNA sequences are listed in Table 1. The RNA sequence CCUUCUU of *PA-miR-2-3p-5* was found to be identical to the seed region of the human *miR-1248* family.

## Target gene prediction using small RNA fragments

Twenty small RNA fragments, including human *miR-1248*, were used as queries in the TargetScan human database to identify potential target genes for these small RNA fragments. TargetScan software scanned the 3′ UTR of human genes that were complementary between the first 2–7 nucleotides of the small RNA fragment. We found that each RNA fragment could target to more than one gene. There were 2877 target genes identified from 16 *P. acnes* small RNA fragments, 35 target genes identified from

| Position 5024-5030 of THSD4 3' UTR | 5' | ...CACGACUCAGUUGACAGAAGGAU... | 3' |
| Position 5466-5472 of THSD4 3' UTR | 5' | ...UCCACAGACAUUUGGAGAAGGAA... | 3' |
| | |                  |||||| | |
| hsa-miR-1248 | 3' | AAAUCGUGUCACGAAUAU**GUUCUUCC**A | 5' |
| EBV-miR-2-3p | 3' | AGAGUCCUAA**GUUCUUCC**UAUU | 5' |
| PA-mir-2-3p-5 | 3' | UAA**GUUCUUCC** | 5' |
| | |        * * * * * * * * | |

**Figure 2** **Alignment of RNA sequences orthologous to the human *THSD4* DNA sequence.** The RNA sequences of *PA-miR-2-3p-5* from *P. acnes* and *EBV-miR-2-3p* from EBOV are shown.

three EBOV miRNAs, and 160 target genes identified from the human *miR-1248* family. Interestingly, three different groups of RNA fragments targeted the *THSD4* gene.

TSPs are a family of multidomain extracellular matrix glycoproteins. The *THSD4* gene has recently been shown to encode TSP type 1 domain-containing 4 in the NCBI database. However, the function of *THSD4* is not yet clear.

## Clustering and conserved segments of human miRNAs

MiRNAs in animals are found in diverse genomic locations. Most miRNAs are encoded in intergenic regions; however, many miRNAs are encoded in the introns of pre-mRNAs or in noncoding RNAs. We first aligned RNA sequences from the microbiome to the human *THSD4* gene, and two conserved segments (AGAAGG) were located in the *THSD4* 3′UTR (Fig. 2). This segment was also the target of the seed region of human *miR-1248* (UCUUCC). Further examination of the microbiome RNA sequences showed that *PA-miR-2-3p-5* and *EBV-miR-2-3p* were present in this segment as well. Moreover, a 6-nt match to the seed sequence (complementary to positions 2–7) was found to retain specificity (*Lewis, Burge & Bartel, 2005*).

## Analysis of THSD4 mRNA expression

To evaluate the importance of the 3′-terminal segments on translational control, HEK293T cells were transfected with the short RNA sequences listed in Fig. 2, and RT-qPCR was used to measure the expression level of *THSD4* mRNA. Notably, cells transfected with *PA-miR-2-3p-5* and *EBV-miR-2-3p* showed similar transcript levels as cells transfected with human *miR-1248* (Fig. 3). Our findings showed that *PA-miR-2-3p-5* may be a very short RNA sequence with only 11 nucleotides but could also behave as an miRNA.

In addition, we also aimed to determine the potential synergistic effects among human and microbiome RNA fragments. The cells were cotransfected with human *miR-1248* plus *PA-miR-2-3p-5*, human *miR-1248* plus *EBV-miR-2-3p*; or *PA-miR-2-3p-5* plus *EBV-miR-2-3p*. Interestingly, we found that dual transfection of the cells resulted in a higher fold change in expression than transfection alone. These findings supported our previous hypothesis that short RNA fragments with a 6-nt match to the seed complementary segment could function as an miRNA. The 3′ UTR of mRNA contained several binding sites for one miRNA, emphasizing the potential importance of synergistic binding of the miRNA to the target. Importantly, TSPs are astrocyte-secreted glycoproteins that can influence

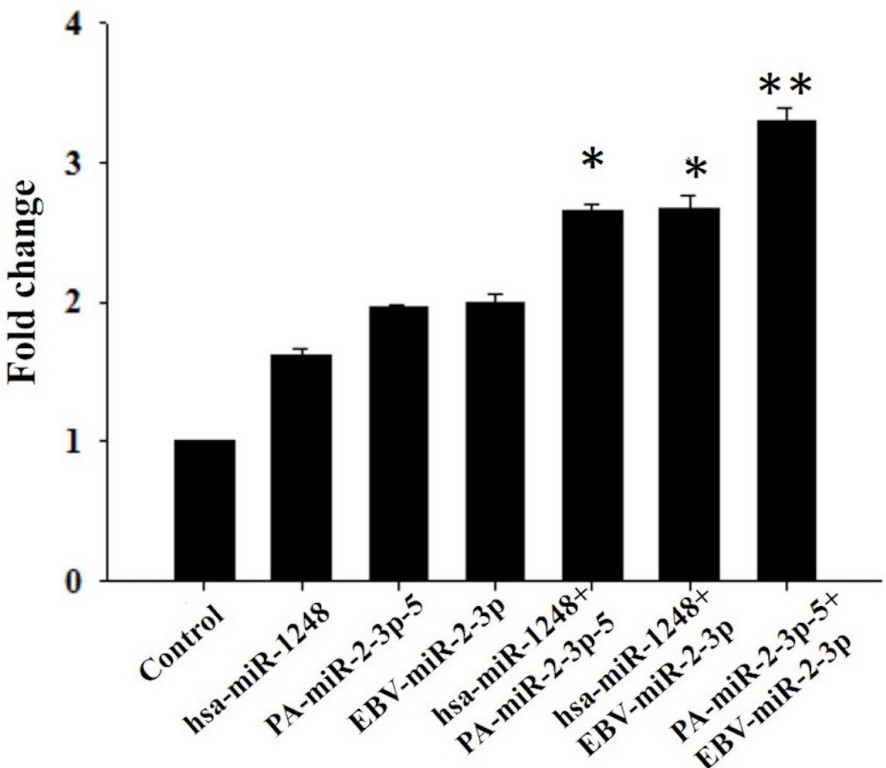

**Figure 3** **Relative expression of *THSD4* mRNA.** The bar graph represents the fold changes in mRNA levels, and the error bars show SEMs ($n = 3$). **$P < 0.001$.

central nervous system (CNS) synaptogenesis (*Wang, Guo & Huang, 2012*). However, the function of the *THSD4* gene has not yet been defined.

Recent studies have shown that EBOV infection could occur through the activities of EBOV miRNAs (*Liang et al., 2014*). As mentioned above, *P. acnes* may play an important role in host protection. Thus, in this study, we established a bioinformatics approach to detect short RNA sequences in *P. acnes* based on sequence alignment; accordingly, our findings demonstrated that these *P. acnes* sequences may act as miRNAs and protect humans from EBOV infection by regulation of the expression of their common target gene *THSD4*.

## ACKNOWLEDGEMENTS

We thank Dr. Eric Huang for thoughtful discussions and ideas. We are also thankful for the encouragement and advice of Dr. Chao Kun-Mao and for the support of our other colleagues. We thank Professor Kang of the Graduate Institute of Toxicology of National Taiwan University College of Medicine for providing HEK293T cells.

## Funding

The authors received no funding for this work.

## Competing Interests

The authors declare there are no competing interests.

## Author Contributions

- Pei-Chun Hsu analyzed the data, contributed reagents/materials/analysis tools, wrote the paper, prepared figures and/or tables, reviewed drafts of the paper.
- Bin-Hao Chiou performed the experiments.
- Chun-Ming Huang conceived and designed the experiments.

## Data Availability

The raw data is included in Table 1 and Fig. 3. The target genes of EBOV-encoded miRNAs and miRNA sequence can be found at https://static-content.springer.com/esm/art%3A10.1007%2Fs11427-014-4759-2/MediaObjects/11427_2014_4759_MOESM1_ESM.pdf. The whole genome of *Propionibacterium acnes* KPA171202 was retrieved from http://www.ncbi.nlm.nih.gov/nuccore/NC_006085.1.

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
