# Peer review of "On revealing the gene targets of Ebola virus microRNAs involved in the human skin microbiome"

_PeerJ, doi:10.7717/peerj.4138_

## Round 0.1 · original submission · Major Revisions

Dear Dr. Hsu,

The study is very important and I thank you for your manuscript. Please see the comments from the reviewers below. I personally recommend to get your manuscript checked by an expert in English language before re-submission. The manuscript has lost the main focus of the study due to difficult to follow language and grammatical errors. However, I feel with substantial corrections and addressing the reviewers concerns, this manuscript can be resent for review.

Best wishes

Reviewer 1 ·

Basic reporting

The article has to be rewritten mainly for the language.
The approach is novel yet there are a few important lacunae, the most obvious being how does the work relate to African population where Ebola is prevalent. Are there any studies on the skin biome from African regions where Ebola is Endemic? How often is Ebola transmitted through skin contact?

Experimental design

Use of different strains of the bacteria may further help to expand the study and verify if microRNAs are indeed important.

Are any other proteins closely associated with the pathological process that could be identified through this approach? For example any other cytokine?

Validity of the findings

The findings look interesting and a little more detail study explaining why macrophages and not DCs maybe relevant in this case needs to be supported.The skin is home to the Langerhans cells which are mostly associated with the immune response which the authors have not described in detail.

Additional comments

Please check for grammar and spelling

Reviewer 2 ·

Basic reporting

Overall, the writing is clear. The structure of the paper is complete except that a conclusion or discussion is missing.

Experimental design

The scientific question is novel. More evidence and discussions need to be provided to make a valid conclusion. See below.

Validity of the findings

The experimental data is controlled. However, in Figure 3, the p-value has not enough power for it was calculated using only 3 data points.

Additional comments

This paper discovers short RNA sequences in P. acnes that might act as microRNA to regulate THSD4. It also found that the short RNA sequence PA-mir-2-3p-5 from P. acnes and micro EBV-miR-2-3p from Ebola can synergistically regulate this gene, through which they claim that these short RNA sequences may protect human from evasion of Ebola. The idea of this paper is novel and very interesting. However, I have several major concerns:

1. Although the short RNA sequence PA-mir-2-3p-5 from P. acnes and microRNA EBV-miR-2-3p from EBOV can synergistically regulate THSD4, the roles of those short RNA sequences in protecting human from evasion of EBOV is not clear. More evidence should be provided to show how P. acnes may prevent EBOV from evasion of the human.

2. It is not clear that increasing of the expression of the gene THSD4 play an important role in the human immune system. In Figure 3, both PA-mir-2-3p-5 and EBV-miR-2-3p can upregulate the expression of THSD4 mRNA. However, the functionality of P. acnes and EBOV are completely different. Please provide some discussion on the possible functionality of THSD4.

3. It would be helpful to explain the reason of using P. acnes in this study. Is it possible for other bacteria, rather than P. acnes, to play an important role in preventing EBOV from evasion of human?

Minor comments:

1. Several grammar errors:

line 83-84, the sentence is composed by two sentences. Thus there
should be a dot in the middle, not a comma

line 123, should be a dot before TargetScan

line 145, should be a comma after 48 hr

line 155, missing subject before “provide"

line 181, that are complementary

line 190, should be “is” not “has”

line 221, should be a comma after "control"


2. Typos

line 130, nucleotides 2-7

line 109, the font of the capital T is different.

Table 1, last row, Human[40], where is [40] from?

3. line 91-92. Thus, miRNAs constitute one of the most abundant classes of gene regulatory molecules in animals. The abundance of miRNA in human is not necessarily applied to in other animals. Please provide other evidence to support this statement.

---

## Round 0.2 · Minor Revisions

Dear Dr. Hsu,

Please consider the reviewers comments carefully. It is important to make the conclusion as clear as possible, supported by data evidence. the readers should be able to clearly understand the hypothetical statements to that of evidence based. I really appreciate you considering these and reworking on the manuscript.

I will look forward to receiving your revised version for possible publication in PeerJ.

Reviewer 1 ·

Basic reporting

No comment

Experimental design

No comment

Validity of the findings

Transmission is through body fluids or broken skin indicating contact with the blood/lymphatic system.So again the question remains how biome on intact skin are related to Ebola transmission.

Additional comments

The role of Langerhans Cells seems to be vital for transmission of the virus through skin. So how do micrbiomes affect the Mobility of LCs?

Reviewer 2 ·

Basic reporting

The conclusion was added at the end of the result section.

Experimental design

no comment

Validity of the findings

no comment

Additional comments

First of all, thank the authors for the revision. The paper is much easier to read now.

The conclusion is still a little confusing that "our findings demonstrated that these P. acnes sequences may act as miRNAs and protect humans from EBOV infection by regulation of the expression of their common target gene THSD4.". The first half of the conclusion is derived based on the results in Figure 2. But the second half of the conclusion is still not supported. By what kind of mechanism that P. acnes prevent EBOV from evasion of the human with the help of THSD4? Since this is the key finding in the paper, please provide rigorous investigation. Otherwise, please limit the statement to supporting results alone.

---

## Round 0.3 · accepted · Accept

Dear Dr. Pei-Chun

Thank you for diligently addressing the concerns of the reviewers. I see potential in the work and of larger interest to the scientific community. I am happy to recommend your manuscript for publication in PeerJ.